# Beyond N Staging in Breast Cancer: Importance of MRI and Ultrasound-based Imaging

**DOI:** 10.3390/cancers14174270

**Published:** 2022-08-31

**Authors:** Valerio Di Paola, Giorgio Mazzotta, Vincenza Pignatelli, Enida Bufi, Anna D’Angelo, Marco Conti, Camilla Panico, Vincenzo Fiorentino, Francesco Pierconti, Fleur Kilburn-Toppin, Paolo Belli, Riccardo Manfredi

**Affiliations:** 1Department of Bioimaging, Radiation Oncology and Hematology, UOC of Radiologia, Fondazione Policlinico Universitario A. Gemelli IRCSS, Largo A. Gemelli 8, 00168 Rome, Italy; 2Institute of Radiology, Catholic University of the Sacred Heart, Largo A. Gemelli 8, 00168 Rome, Italy; 3Institute of Pathology, Università Cattolica del S. Cuore, Fondazione Policlinico “A. Gemelli”, 00168 Rome, Italy; 4Cambridge Breast Unit, Cambridge University Hospital NHS Foundation Trust, Addenbrookes’ Hospital, Hills Road, Cambridge CB2 0QQ, UK

**Keywords:** MRI, ultrasound, N-stage, staging, breast cancer, axillary lymph node, sentinel lymph node biopsy, axillary lymph node dissection

## Abstract

**Simple Summary:**

Breast cancer is the most frequent cancer affecting women and metastatic breast cancer is still the leading cause of death from all cancers in women, accounting for about 3.6% of all deaths in women. The N-stage represents the main prognostic factor affecting the rate of recurrence and the therapeutic management so that a correct staging of the axillary lymph node status is fundamental. Since clinical examination of the axillary cavity is associated with a high false negative rate, reaching values of 45%, the role of imaging becomes crucial to obtain an accurate assessment of loco-regional lymph nodes at the time of diagnosis. In this setting, Ultrasound and Magnetic Resonance Imaging (MRI) represent two important diagnostic tools. In particular, MRI represents an accurate and reproducible technique, which allows an accurate staging of the “N-stage”.

**Abstract:**

The correct N-staging in breast cancer is crucial to tailor treatment and stratify the prognosis. N-staging is based on the number and the localization of suspicious regional nodes on physical examination and/or imaging. Since clinical examination of the axillary cavity is associated with a high false negative rate, imaging modalities play a central role. In the presence of a T1 or T2 tumor and 0–2 suspicious nodes, on imaging at the axillary level I or II, a patient should undergo sentinel lymph node biopsy (SLNB), whereas in the presence of three or more suspicious nodes at the axillary level I or II confirmed by biopsy, they should undergo axillary lymph node dissection (ALND) or neoadjuvant chemotherapy according to a multidisciplinary approach, as well as in the case of internal mammary, supraclavicular, or level III axillary involved lymph nodes. In this scenario, radiological assessment of lymph nodes at the time of diagnosis must be accurate. False positives may preclude a sentinel lymph node in an otherwise eligible woman; in contrast, false negatives may lead to an unnecessary SLNB and the need for a second surgical procedure. In this review, we aim to describe the anatomy of the axilla and breast regional lymph node, and their diagnostic features to discriminate between normal and pathological nodes at Ultrasound (US) and Magnetic Resonance Imaging (MRI). Moreover, the technical aspects, the advantage and limitations of MRI versus US, and the possible future perspectives are also analyzed, through the analysis of the recent literature.

## 1. Introduction

Breast cancer is the most frequent cancer affecting women worldwide in 2020, accounting for 24.5% of all female-diagnosed cancers, and represents the leading cause of death from all cancers in women, accounting for the 15.5% of all deaths [1,2].

Axillary lymph node (ALN) status is the main prognostic factor that influences the rate of recurrence and the therapeutic management [3,4], so that accurate staging of the ALNs is fundamental.

Traditionally, the axillary staging was surgically assessed by axillary lymph node dissection (ALND), which is potentially associated with complications such as uncomfortable postoperative drains and seroma, cellulitis, lymphedema, range-of-motion restriction, arm paresthesia, and pain. Many of the patients who underwent this surgery had no nodal metastases and suffered the morbidities without an oncologic benefit [5]. 

Sentinel lymph node biopsy (SLNB) is a minimally invasive surgical method for axillary staging in patients with primary breast cancer with lower morbidity and better quality of life [6]. The principle behind the sentinel lymph node concept is based on the entry of cancer cells into the lymphatic system, followed by an orderly progression first to the sentinel lymph nodes and only then to the remaining lymph node stations. 

Since clinical examination of the axillary cavity is associated with a high false negative rate, reaching values of 45% [7], the role of imaging becomes crucial to identify lymph nodes with or without suspicious features, to predict the pathological state of the lymph nodes and to direct the diagnostic and therapeutic process towards SLNB, ALND, or neoadjuvant chemotherapy (Figure 1). 

SLNB is the standard surgical approach to axillary staging in patients with clinically node-negative breast cancer. National Comprehensive Cancer Network (NCCN) guidelines have recently expanded the indications for SLNB, suggesting the procedure also for patients with T1 or T2 stage, clinically node negative and from 0 to 2 suspicious axillary level I or II lymph nodes at imaging [8]. If no sentinel lymph node metastases are detected, ALND should not be recommended. Historically, axillary lymphadenectomy was indicated in case of a positive lymph node on pathological examination. Nevertheless, the scenario has further changed since the results of the American College of Surgeons Oncology Group (ACOSOG) Z0011 trial [9,10]. If specific criteria are respected (Table 1), women with early-stage breast cancer who show one or two lymph node metastases on pathologic examination after SLNB, should not undergo ALND [11]. 

The patients selected according these criteria showed a non-inferior to overall survival for those treated with ALND, without suffering the complications associated with ALND. 

SLNB is not indicated in case of 3 or more positive axillary level I or II lymph nodes or in case of level III axillary or internal mammary or supraclavicular lymph nodes on physical examination and/or imaging confirmed by biopsy. In these cases, ALND or neoadjuvant chemotherapy should be performed according to a multidisciplinary approach [12].

In this scenario, radiological assessment of lymph nodes at the time of diagnosis must be accurate. False positives may preclude sentinel lymph node in an otherwise eligible woman [13]. In contrast, false negatives may lead to an unnecessary SLNB and the need for a second surgical procedure [14].

Radiologic staging has also become more important as the use of neoadjuvant chemotherapy (NAC) has increased. NAC followed by surgery is the standard treatment for patients with clinically node-positive breast cancer [15], showing several potential advantages. In particular, it can potentially result in downstaging of tumor and lymph node involvement, respectively ensuring breast-conserving surgery in patients who would have needed mastectomy or obviating the need for ALN dissection [16]. Imaging can be performed before and during neoadjuvant chemotherapy to assess treatment efficacy in order to tailor the chemotherapy regimen [17], changing the treatment approach if the tumor is not responding [16] or allowing a more conservative treatment in case of downstaging of the axillary nodal involvement.

Therefore, the accurate knowledge of regional breast lymph node anatomy and imaging features of normal and abnormal lymph nodes is essential to determine the cancer stage. Moreover, the radiologist should know the potentialities and limitations of the main imaging techniques used to evaluate lymph node involvement in breast cancer. 

## 2. Anatomy of Axilla and Regional Lymph Nodes: The N-Stage

The axilla is composed of the axillary artery and vein, brachial plexus, lymph nodes, fat tissue, accessory breast tissue, skin, and subcutaneous glands. Regional nodes for the breast include axillary, supraclavicular, and internal mammary nodal chains. 

The margins of the pectoralis minor muscle divide the axilla into levels I, II, and III. Level I axillary nodes are located inferior and lateral to the lateral border of the pectoralis minor muscle. Level II axillary nodes are located posterior to the pectoralis minor muscle and include interpectoral Rotter nodes (nodes located between the pectoralis minor and pectoralis major) [18]. Level III axillary nodes (infraclavicular) are located superior and medial to the medial border of the pectoralis minor muscle. Figure 2 and Figure 3 represents the three axillary levels as they appear both in Ultrasound (Figure 2) and MRI Images (Figure 3).

Supraclavicular nodes are located superior to the clavicle and lateral to the internal jugular vein [19]. Internal mammary nodes are located along the parasternal intercostal spaces, running along the course of the internal mammary artery and vein between the pleura/endothoracic fascia and the chest wall margin. 

LNs drainage generally proceeds in an ordinally manner from level I to level II, to level III, and finally into the thorax [20]. According to the eighth edition of the American Joint Committee on Cancer (AJCC) TNM classification, axillary lymph node involvement in level I or II indicates N1 stage, whereas lymph node metastases in level III indicate N3a stage. Level III nodes drain into supraclavicular nodes so that metastasis to this level indicates N3c stage. Isolated metastases to the internal mammary nodes indicate N2 stage. This condition occurs in 1–5% of breast cancers and usually nodes metastases come from deep or medial lesions. Association between internal mammary nodes and level I or II axillary nodes indicates N3b stage. The N3 stage automatically indicates IIIC stage, independently from T-stage.

Normal lymph nodes (LN) have an oval or reniform shape and they are formed by a capsule that surrounds three compartments: cortex, paracortex, and medulla [21] (Figure 4).

The capsule deepens in the central area to constitute the fatty hilum. Lymphatic flow enters the cortex through the afferent channel in the marginal sine and then exits the hilum through the efferent channel. In addition, lymph nodes receive a specific blood supply through arterial and venous branches. As tumor cells deposit and coalesce near the marginal sinus, the cortex progressively thickens [12,22]. 

## 3. Ultrasound

Ultrasound (US) is normally the first modality of imaging to evaluate axillary lymph nodes. Axillary US should be performed using a high-frequency (11–15-MHz) linear-array transducer. In the case of patients with a large axillary fat pad, a lower frequency setting (5–7.5-MHz) can be used. The patient is placed in the supine position, with her hand behind her head so that the arm is abducted and externally rotated (“ABER” or “bathing beauty” position) [23].

Level I lymph nodes should be systematically researched by scanning along the axillary vessels and exploring the axillary fatty tissue. It is also very important to scan inferiorly, down through the axillary tail, because abnormal nodes are frequently found there [14]. Levels II and III are can also be scanned to detect enlarged nodes. 

Scanning the supraclavicular area and along the lateral margin of the sternum may be useful to detect supraclavicular or internal mammary lymphadenopathy, respectively.

Evaluation of lymph nodes should include size, general shape, cortical thickness, hilum appearance, and vascularization with the color Doppler technique [24]. 

The normal axillary lymph node should be oval with smooth and well-defined edges. The cortex should be slightly hypoechoic and uniformly thin, measuring 3 mm or less. The fatty hilum should be preserved with an echogenic aspect, and it should constitute the majority of the node. Color Doppler imaging demonstrates arterial flow in the hilum [14,25].

Breast metastases generally enter the node through an afferent lymphatics and then deposit in the subcapsular sinusoids.

According to the AJCC metastatic deposits measuring less than 0.2 mm are called “isolated tumor cells”, and deposits between 0.2 and 2.0 mm are defined as “micrometastases”. This level of disease is not identifiable at imaging [14], but micrometastasis and isolated tumor cells do not affect general survival. Therefore, the sensitivity of their detection is of less importance [26].

Growth in the marginal sinus can result in a focal cortical bulge or eccentric cortical thickening detectable on imaging [14]. 

The earliest morphologic sign of the presence of a metastatic node is the diffuse thickening (>3 mm) or focal bulge of the cortex. A true abnormal cortical bulge is seen as focal thickening of the cortex that does not follow the margin of the hyperechoic hilum and should be distinctly hypoechoic. Diffuse thickening of the cortex can be a pitfall sign because it can also be seen in reactive nodes. Eccentric cortical thickening is slightly more suspicious than diffuse thickening, but these criteria are difficult to apply and have a low positive predictive value (PPV) because they are non-specific.

Later, tumor cells spread from the cortex into the deeper nodal parenchyma, proliferating in a heterogeneous fashion and replacing the normal architecture, resulting in the destruction of the intranodal “angioarchitecture”. This distortion can lead to engorgement of peripheral vascularity that combined with subcapsular metastasis may be the cause of the nonhilar cortical blood flow (NHBF) seen at color Doppler US [14]. NHBF is the appearance of peripheral vascular flow at the cortex of the node with no detectable connection to the hilum. This finding has been shown to have a high PPV (78%) for metastasis in the setting of ipsilateral invasive breast cancer. When an abnormal cortical bulge is seen associated with NHBF it becomes more specific. 

After continued growth, metastases can obliterate the histologic features of the node and replace the entire node. The imaging finding that reflects this process is the appearance of a uniform rounded hypoechoic node with loss of the fatty nodal hilum. These findings have a PPV of 58%. In these cases, performance of US-FNA or USCNB is recommended for these nodes.

The ultrasonographic features of pathological nodes are summarized in Figure 5.

Extranodal spread of tumor into the adjacent axillary fat can occur. At imaging, microscopic extranodal deposits can cause the loss of definition of the cortical margin or the appearance of spiculation. Microscopic extranodal deposits can also stimulate the growth of perinodal neovascularity, which is perhaps another source of abnormal NHBF detected at US. Ultimately, the node may be totally replaced by an irregular mass. Occasionally, microcalcifications can be seen in the node at US.

## 4. Magnetic Resonance Imaging (MRI)

Morphological evaluation of axillary nodes is also performed at Magnetic Resonance Imaging (MRI), which represents an important technique in the staging of breast cancer, including the detection of pathological nodes. 

Normal nodes are characterized by a hypointense signal on T1-weighted images and intermediate to increased signal on T2-weighted and inversion recovery sequences. In the non-fat-saturation sequence the hilar fat has increased signal intensity, while in the fat-saturation sequence the hilum appears hypointense. Nodes enhance rapidly and homogeneously at dynamic contrast-enhanced (DCE)—MR Imaging. 

Morphologic features which can suggest pathological nodes are represented by cortical thickening or cortical irregularity, spiculation, effacement of fatty hilum, heterogeneous enhancement, and round shape or a long axis to short axis ratio of less than 2 (Figure 6).

Some authors did find that nodes with a cortical thickness of less than 3 mm had a high NPV (91%) for metastatic involvement. Even if both benign and malignant nodes enhance intensely on DCE-MR images, it has been demonstrated that the lower enhancement of nodes has a high NPV (100%) for metastasis [14]. 

Two other MRI features which have potential diagnostic utility when present are “perifocal edema” and “rim enhancement”. Perifocal edema is defined as the presence of areas with marked T2 prolongation in the fat surrounding a node, while rim enhancement is defined as a higher signal intensity at the periphery of a node than at its center on DCE-MR images obtained 11 min after infusion. Both of these findings had a 100% PPV for the detection of metastases in the study by Baltzer et al. [27].

An important advantage of MRI is the likelihood to obtain a large view of the axillae. This allows to explore the deeper axillary node levels and also to compare right and left axillary nodes in terms of number, size, and morphology. In Baltzer et al. series, positive “asymmetry” and “irregular margin” was a significant predictor for nodal positive (PPV 100%), whereas “symmetry” and “homogeneous cortex” was highly predictive of nodal negative (NPV 94.3%) for excluding metastasis [27]. 

In another series, the asymmetry of LNs in terms of number or size compared with the contralateral side was confirmed as an additional finding suggestive of metastasis [25]. 

Diffusion Weighted Imaging (DWI) can improve the detection of pathological axillary nodes. They appear hyperintense on high b-value images (at least b-800 s/mm^2^) and hypointense on the Apparent Diffusion Coefficient (ADC) map (Figure 7). 

It was observed a statistically significant difference between the Apparent Diffusion Coefficient (ADC) values on DWI Imaging of pathological and non-pathological lymph nodes, being lower in the pathological ones. The possible functional mechanism is that high cellularity in lymph node metastases produced by the tumor cells, may decrease extracellular and intracellular spaces, and restrict water molecules’ mobility, finally leading to a reduction in ADC value [28]. Fardanesh et al. have found that the median values of mean ADC, maximum (max) ADC and minimum (min) ADC were significantly lower for malignant lymph nodes rather than benign axillary lymph nodes. In their study the ADC values (×10^−3^ mm^2^/s) of benign axillary lymph nodes ranged from 0.522 to 2.712 for mean ADC, from 0.774 to 3.382 for max ADC, and from 0.071 to 2.409 for min ADC; the ADC values (×10^−3^ mm^2^/s) of malignant axillary lymph nodes ranged from 0.796 to 1.080 for mean ADC, from 1.168 to 1.592 for max ADC, and from 0.351 to 0.688 for min ADC [29]. However, while these differences are significant in Fardanesh et al. series (*p* < 0.001), the range of possible ADC values for benign and malignant axillary lymph nodes can overlap with the range of ADC values reported in other studies in the literature. This overlap can represent a diagnostic challenge. The least overlap occurs with mean ADC, suggesting that mean ADC is the most useful ADC parameter for differentiating between benign and malignant axillary lymph nodes. According to Fardanesh et al. the mean ADC threshold that resulted in the highest diagnostic accuracy for differentiating between benign and malignant lymph nodes was 1.004 × 10^−3^ mm^2^/s, yielding an accuracy of 75%, sensitivity of 71%, specificity of 79%, the PPV of 77% and negative predictive value (NPV) of 74%. This mean ADC threshold is lower than the mean ADC threshold of 1.300 × 10^−3^ mm^2^/s recently recommended by the European Society of Breast Imaging (EUSOBI) for breast tumors but not for axillary lymph nodes. 

Intravoxel incoherent motion (IVIM) MRI is another DWI technique for the assessment of lymph node metastasis. IVIM is sensitive to molecular diffusion and the random flow of blood in capillaries to characterize tissue cellularity and vascularity. Liu et al. found that one IVIM feature, the pseudo-diffusion coefficient (D*), related to the dynamic flow rate of microvasculature is significantly associated with metastasis, with a sensitivity of 73.5% and a specificity of 84% [30]. 

The role of MRI dedicated axillary sequence for the detection of axillary lymph node metastasis is controversial. The dedicated axillary sequence can eliminate pulsation artifact due to the use of phase encoding direction and can be used with the same breast coil for the primary breast cancer or using a separate surface coil on the patient’s axilla [31]. With the surface coil, the complete axillary region is visualized in a coronal plane. Dedicated axillary MRI exams may improve the visualization of lymph nodes in axillary levels II and III which may not be identified easily due to the location [32]. However, several studies have not found a significant advantage including specific axillary sequences in the MRI protocol. Ha et al. have concluded that standard MR and dedicated axillary MRI sequences demonstrate similar diagnostic performance to differentiate positive and negative axillary lymph nodes with 64.7 and 66.2% sensitivity, 94.0 and 93.3% specificity, and 94.3 and 94.4% NPV, respectively [33]. 

In conclusion, the diagnostic performance of T2-weighted standard breast MRI with complete Field of Interest (FOV) of the axillary region is comparable with that of the T2-weighted dedicated axillary MRI regarding the assessment of node breast cancer. Therefore, optimization of T2-weighted standard breast MRI protocol by including a complete FOV of the axillary region can be recommended in clinical practice.

Because of the increase in the use of MRI it has been developed a faster protocol: abbreviated MRI forfeits the conventional delayed-phase kinetic information by eliminating the later postcontrast sequences (second and third postcontrast time points); the typical imaging time is less than 10 min. Abbreviated MRI should include a localizer acquisition and one precontrast and one postcontrast axial gradient-echo acquisition with an in-plane resolution of 1 mm or less and section thickness of 3 mm or less, with or without a T2-weighted sequence. Ultrafast MRI is not necessarily a component of abbreviated MRI but is a novel sequence developed to capture early contrast material wash-in at high temporal resolution (typically ≤6–7 s). An ultrafast sequence allows rapid sequential imaging within the first 2 min after contrast material injection to render an early wash-in kinetic curve, as opposed to the conventional delayed washout kinetic curve. In fact, conventional dynamic contrast-enhanced MRI typically acquires at least three postcontrast series to generate a delayed kinetic curve, with the first post-contrast time point at 90–120 s. A washout delayed kinetic curve is highly associated with malignancy. The initial contrast material wash-in kinetics (within the first 120 s), which are not available at conventional MRI, can now be acquired with ultrafast MRI [34].

For the evaluation of lymph nodes, a specific contrast agent for MRI has been developed to detect malignant nodal infiltration independently of lymph node size. This MR contrast agent is classified as a nanoparticle (mean diameter, 30 nm) and is composed of an iron oxide core coated with low-molecular weight dextran. The class of these MR contrast agents is collectively known as ultrasmall superparamagnetic iron oxide (USPIO). After intravenous USPIO administration, non-metastatic lymph nodes generally show uptake of contrast material, which results in decreased signal intensity (SI) on T2- and T2*-weighted MR images, whereas metastatic lymph nodes generally do not exhibit any uptake, and SI remains unchanged. This stability in SI within metastatic lymph nodes has been attributed to the replacement of macrophages by metastatic cells, which lack the reticuloendothelial activity and show no USPIO uptake. The study of Memarsadeghi et al. confirms the value of USPIO-enhanced MRI as a potential diagnostic tool, on the basis of enhancement patterns, for preoperative nodal staging in patients with breast carcinomas [35]. 

Preliminary studies are encouraging, with a sensitivity and specificity value of up to 98% and 93%, respectively, but most of the study have been small and further studies are needed. Currently, USPIO administration in clinical practice is poorly widespread. In the United States it is not yet approved.

## 5. US versus MRI

US is the usual first line modality for assessment of the axilla. Advantages of US include low cost, widespread availability, and the possibility to serve as a guide for nodal biopsy [36]. Another huge benefit of axillary US over MRI is the ability to insert a marker clip into the pathological nodes. This procedure is crucial for targeted axillary dissection, which allows reducing the post-surgical morbidity, particularly in patients who have managed to be downstaged after chemotherapy.

Ultrasonography is the modality that allows for better defining of the morphology of the lymph node, being able to represent cortical morphologic change better than MRI, due to the higher resolution of US compared to MRI [37]. However, the accuracy of US is variable because it is operator-dependent and its performance may be hampered by the patient body habitus.

The preoperative axillary US shows a high NPV of about 80% [38]. When based on morphologic criteria, US can be highly specific when used by experienced operators, showing a specificity of 88–98% and a wide range sensitivity of between 26% and 76% in detecting non-palpable lymph node metastases [39].

If lymph node size was used as a criterion for US positivity, sensitivity ranged from 49% to 87% and specificity from 55.6% to 97.3% [39]. US accuracy can be improved by US-guided lymph node sampling, such as fine-needle aspiration (US-FNA) and central-needle biopsy (US-CNB). A meta-analysis showed that both US-CNB and US-FNA had a specificity of 100%, but US-CNB was superior to US-FNA in diagnosing axillary lymph node metastases with a sensitivity of 88% versus 74%, respectively [40].

Compared to ultrasonography, MRI provides a more global view of the axilla and it can be used for the nodes evaluation as well as for the simultaneous local staging, assessing the extension of the breast cancer and affected area such as the skin and pectoral muscle, with a better definition of prognosis [41]. In addition, MRI allows a comparative evaluation of the two sides even in larger patients [27] and a better evaluation of the internal mammary and supraclavicular lymph nodes [42]. Moreover, MRI is less operator-dependent. However, it is associated with higher costs and the presence of pulsation artifact from the heart can occasionally worsen the evaluation of ALNs, especially at levels II and III [36]. 

In a meta-analysis of 21 eligible studies to evaluate the efficacy of MRI, the pooled sensitivity and specificity of MRI were 82% and 93%, respectively [43]. T2-weighted MRI for axillary lymph node staging has been shown to be highly specific, with a specificity of 92.3%, a sensitivity of 50.0–62.5%, a PPV of 57.1–62.5%, and a NPV of 90.0–92.3% [32]. Baltzer et al. reported that the presence of perifocal edema in T2-weighted sequences is associated with the highest PPV for malignancy of 100% [27]. The use of dedicated T2-weighted axillary MRI may further increase diagnostic accuracy; however, NPV is still insufficient (86–89%) to exclude axillary metastases in patients with breast cancer, so even with dedicated protocols a negative MRI does not rule out metastatic lymph node disease [44]. The diagnostic accuracy for axial T1-weighted MRI and DWI were 85% and 80%, respectively [45]; the addition of DWI can improve sensitivity at the expense of lower specificity [44]. Contrast-enhanced breast MRI has been shown to be more accurate than non-contrast breast MRI in predicting pathologic positivity of axillary lymph nodes [46].

The publication of the ACOSOG Z0011 Trial changed the role of imaging in the assessment of lymph node status in patients with clinically negative lymph nodes. Historically, these patients underwent SLNB and, in case of a positive lymph node on pathological examination, axillary lymphadenectomy was indicated. Therefore, imaging was limited to attempting to predict the presence or absence of lymph node metastasis on histopathological examination. In the post-ACOSOG Z0011 trial era if lymph node metastases are observed after SLNB, ALND is still not indicated if lymph node metastases are limited to 1 or 2 (“low nodal burden”). In contrast, the presence of three or more lymph node metastases on histopathologic analysis defines a “high nodal burden” and this directs the choice toward axillary lymphadenectomy or neoadjuvant chemotherapy. 

Consequently, imaging should not only predict the presence of lymph node metastases but should predict high lymph node burden preoperatively [37]. In these cases, an ALND could be performed directly, without the need for SLNB.

MRI can integrate US-guided lymph node sampling in predicting a high tumor burden. Kim et al. demonstrated that a higher (≥2) T stage (OR = 5.17) and a higher number of suspicious ALNs (2 suspicious ALNs, OR = 69.00; ≥3 suspicious ALNs, OR = 93.55) were independently associated with high lymph node burden. This study defines axillary lymph nodes as suspicious when one or more findings were noted as follows: cortical thickness > 2 mm and presence of eccentric cortical thickening or loss of fatty hilum [37].

## 6. Future Perspectives

Radiological assessment of breast nodes may be hindered by variations in the technique, as well as by interobserver and intraobserver variability in interpretation. Multiple investigators have developed computer vision and machine learning methods for computer-aided diagnosis and the quantitative characterization of breast lesions and axillary nodes on clinical images. Radiomics represents the extraction of a large amount of quantitative information called “features” from medical images (such as CT, MRI, or PET) through the process of automatic segmentation of the pathological findings, which are then converted into quantifiable data and analyzed by software. The so provided computer-extracted data can be related to tumor biology and other clinical, pathological, and genomic data. The process of feature extraction can be implemented by Artificial Intelligence (AI) system. This consists of automatic segmentation software, which has the additional potential to obtain a more detailed analysis of MRI without increasing the time burden on the interpreting radiologist. It is important to note, however, that the AI system studied here has the potential to help radiologists reduce classification errors but not detection errors because the radiologist must first identify the lesion to make use of the AI system [47].

In a recent systematic review [48] which analyzed the performance of radiomic assessment of axillary nodes from breast cancer, the best-reported accuracy value of radiomic models was 89% in the Cui et al. Series [49]. In this work, the combination of morphological and texture features had the highest performance. In addition, a nomogram that scored morphological and texture features to calculate the probability of axillary lymph node metastases (ALNM) was established. 

Other published studies explored the association between breast cancer and ALNM. Chai et al. demonstrated that among the different MRI sequences, the combination of the second post-contrast phase features and kinetic features had the highest performance and that the presurgical radiomic signatures of primary breast tumor were associated with ALNM [50]. Liu et al. found that radiomic analysis based on dynamic contrast-enhanced MRI predicted ALNM with an accuracy of 0.85 (AUC of 0.83) [51]. Similar results (AUC of 0.86) were shown by Shan et al., who validated a nomogram model to detect ALNM in patients with invasive breast cancer, incorporating the kinetic curve model and five radiomic features extracted from DCE-MRI [52].

Dong et al. showed that the integration of different morphological sequences as T2-weighted with fat saturation images combined with DWI images reached similar high predictive performance (AUC: 0.805) [53]. 

Other authors demonstrated that the use of nomograms that included other also clinical parameters showed better accuracy in respect of the radiomic features alone. In particular, Han et al. showed as the nomogram that considered both lymph node radiomic signature and lymph node palpation had a higher performance (AUC: 0.84 vs. 0.78) than the radiomic signature alone in detecting the number of metastatic lymph nodes (less than two positive nodes/more than two positive nodes), with the important implications in selecting patients to ALND or SNLB [54].

Other validated nomograms have been demonstrated able (AUC: up to 90) to successfully stratify patients with early-stage breast cancer according to their risk of ALNM and disease recurrence [55], and to evaluate surgical outcomes [56]. Radiomics and artificial intelligence are very promising new approaches for the prediction of axillary metastases but still limited due to their technical and methodological complexity, and therefore, not yet applied to routine clinical practice.

New prospective studies with larger sample size and focused on the clinical aspect are needed to thoroughly investigate the behavior of breast cancer concerning axillary involvement and their radiomic mirroring. 

## 7. Conclusions

Evaluation of axillary nodes at clinical examination is burdened by a high false negative rate of up to 45%.

US is the usual first line modality for assessment of the axilla, representing the Gold Standard for better defining of the morphology of the lymph node due to its higher resolution compared to MRI. The high negative predictive value allows to select patients with low nodal burden (one or two positive nodes), who are candidate to sentinel lymph nodes biopsy (SLNB), and patients with high nodal burden (more than two positive nodes), who are candidate to axillary lymph nodes dissection (ALND) or chemotherapy. Moreover, US may provide a guide to perform biopsy or to insert a marker clip into nodes to orientate axillary dissection. 

However, the accuracy of US is variable because it is operator-dependent and its performance may be hampered by the patient body habitus, varying from 26% to 76%. In this context MRI can integrate US-guided lymph node sampling in predicting a high tumor burden. Even if MRI has a lower spatial resolution compared to US, it is less-operator dependent, reaching a sensitivity and specificity values up to 82% and 93%, respectively. Moreover, the use of a dedicated T2-weighted axillary coil may further increase the diagnostic accuracy of MRI. Another important advantage of MRI is represented by the possibility to have a global view of the axilla and of the contralateral side to evaluate node symmetry and the simultaneous evaluation of both nodes (N-stage) and primitive tumor (T-stage), with a better definition of prognosis. 

Radiomics and artificial intelligence are very promising new approaches for the prediction of axillary metastases but still limited due to their technical and methodological complexity, and therefore, not yet applied to routine clinical practice.

In conclusion, MRI represents an important diagnostic tool to assess the correct clinical N-stage, which is crucial to select patients for SLNB versus ALND or chemotherapy. 

## Figures and Tables

**Figure 1 cancers-14-04270-f001:**
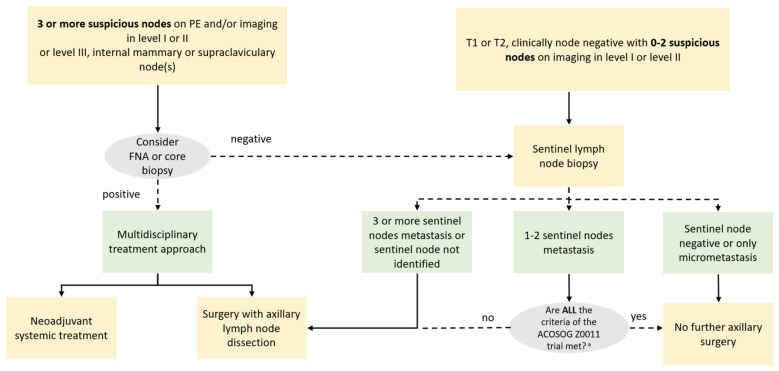
A simplified approach to surgical axillary staging in patients with breast cancer. ^a^ ACOSOG criteria are listed in Table 1. PE: physical exam; ACOSOG: American College of Surgeons Oncology Group.

**Figure 2 cancers-14-04270-f002:**
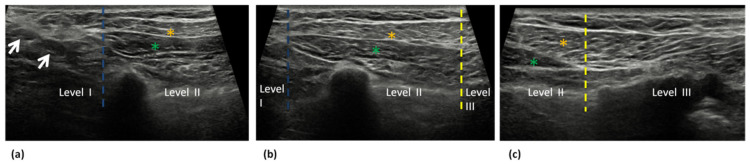
Ultrasound axial images show the axillary lymph node levels I, II, and III. (**a**) Level I is located laterally to the lateral margin (blue dotted line) of the pectoralis minor muscle (green asterisk). Normal axillary level I lymph nodes are also visible (white arrows) (**b**) Level II is located between the lateral (blue dotted line) and the medial margin (yellow dotted line) of the pectoralis minor muscle (green asterisk). (**c**) Level III is located medially to the medial margin (yellow dotted line) of the pectoralis minor muscle. The pectoralis major muscle (orange asterisks) is appreciable superficially to the minor pectoralis muscle in a, b and c.

**Figure 3 cancers-14-04270-f003:**
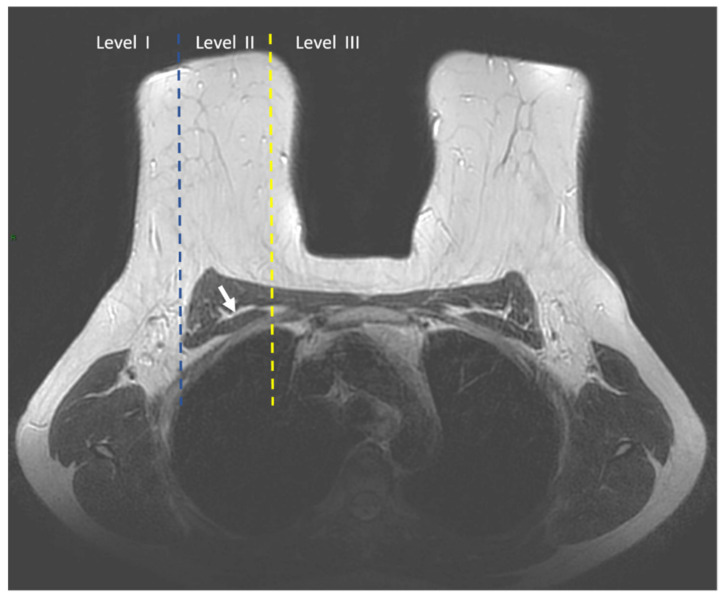
T2-weighted axial image shows the axillary lymph node levels I, II, and III. Level I is located laterally to the lateral margin (blue dotted line) of the pectoralis minor muscle (white arrow), level II is located between the lateral (blue dotted line) and the medial margin (yellow dotted line) of the pectoralis minor muscle, and level III is located medially to the medial margin (yellow dotted line) of the pectoralis minor muscle.

**Figure 4 cancers-14-04270-f004:**
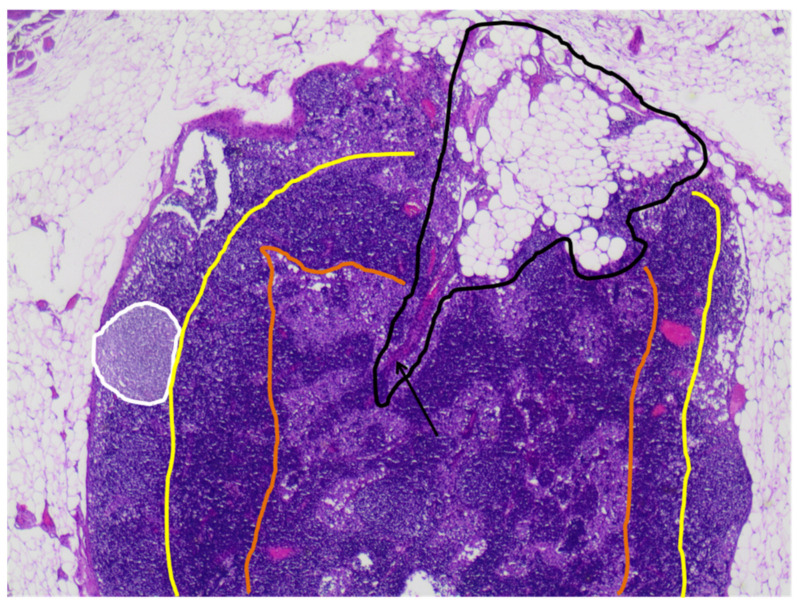
Hematoxylin and eosin-stained slide at 20× magnification shows a normal breast lymph node with the three compartments: cortex, externally to the yellow line, with a follicle inside (white line); paracortex, externally to the orange line; and medulla, internally to the orange line. Inside the lymph node, the fatty hilum (lined by the black line) with a blood vessel (arrow) is also seen.

**Figure 5 cancers-14-04270-f005:**
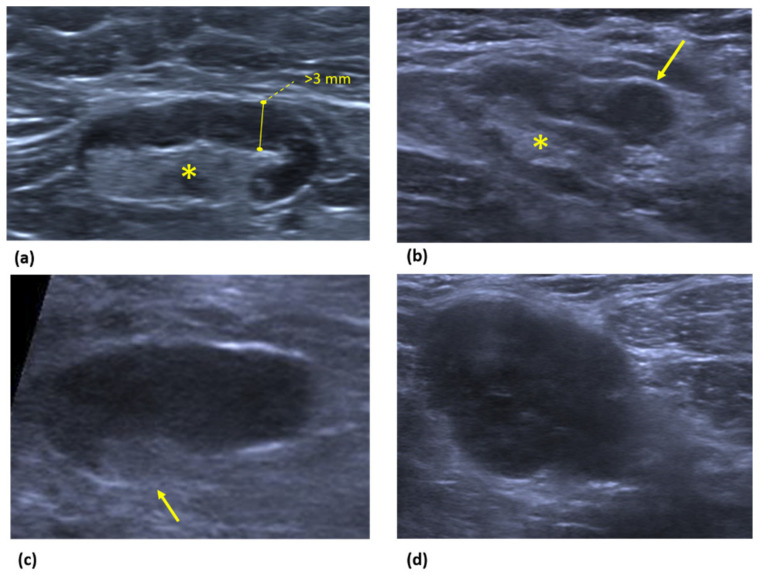
Images show US findings of suspicious axillary lymph nodes. (**a**) A diffuse hypoechoic cortex greater than or equal to 3 mm (solid line) with preserved hilum (asterisk) may be the earliest sign of malignancy, but it is not specific. (**b**) US image shows an axillary lymph node with focal eccentric cortical thickening (arrow) and preserved hilum (asterisk), which represents a more suggestive sign of malignancy. (**c**) US image shows a suspicious node with cortical thickness and displacement of the hilum, which represents a more specific sign (arrow). (**d**) US image demonstrates a roundish hypoechoic lymph node with replaced hilum; a replaced hilum is the most specific feature of malignancy.

**Figure 6 cancers-14-04270-f006:**
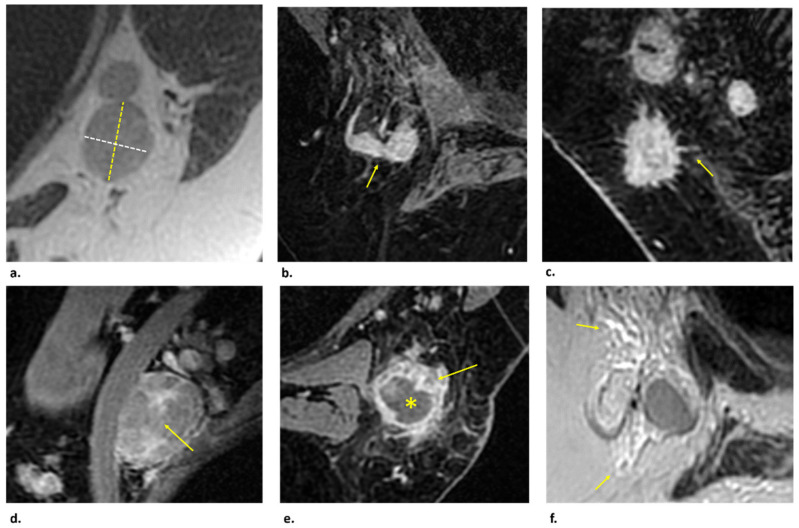
The images show the main features suggestive of lymph node malignancy appreciable at Magnetic Resonance Imaging. (**a**) T2-weighted axial image shows a lymph node with a long axis (dotted yellow line) to short axis (dotted white line) ratio less than 2. (**b**,**c**) Gradient echo (GRE) fat-suppressed (FS) contrast–enhanced T1-weighted axial images show pathological lymph nodes characterized by cortical thickening (arrow in (**b**)) and spiculation (arrow in **c**). (**d**) Sagittal GRE FS contrast–enhanced T1-weighted image demonstrates heterogeneous enhancement of a metastatic axillary lymph node (arrow). (**e**) GRE FS contrast–enhanced T1-weighted axial image shows a restructured lymph node, with a central necrotic component (asterisk) and increased contrast enhancement at the periphery (arrow), also called “rim enhancement”. (**f**) T2-weighted axial image shows the presence of “perifocal edema” defined as areas with marked T2 prolongation in the fat surrounding a node (arrows).

**Figure 7 cancers-14-04270-f007:**
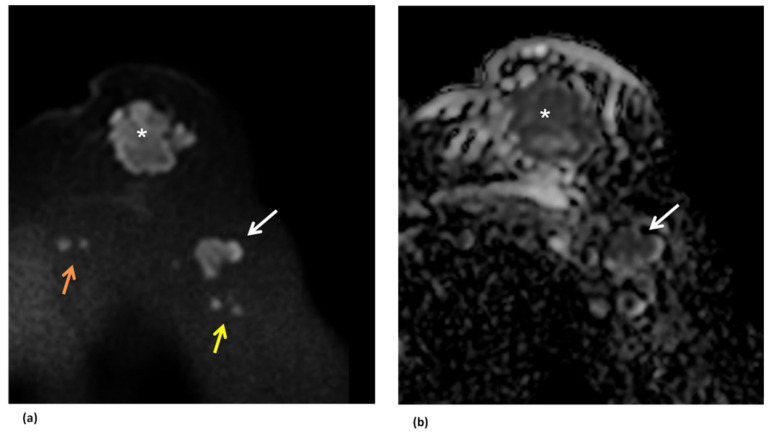
The figure shows pathological axillary lymph node (white arrows) on diffusion-weighted imaging (DWI). They are characterized by high signal intensity on the DWI image at b = 1000 s/mm^2^ (**a**) and low signal intensity on the apparent diffusion coefficient map (**b**). The lymph node has the same signal features of the primitive tumor (white asterisk). Other small lymph pathological nodes are appreciable in the axillary (yellow arrows) and internal mammary (orange arrows) stations.

**Table 1 cancers-14-04270-t001:** The table shows the inclusion criteria of the American College of Surgeons Oncology Group (ACOSOG)—Z0011 Trial [9].

American College of Surgeons Oncology Group (ACOSOG) Criteria—Z0011 Trial
■ T1 or T2 tumor
■ Clinically negative nodes
■ 1 or 2 positive nodes on sentinel lymph node biopsy
■ Planned breast conserving surgery
■ Planned whole-breast radiation therapy
■ No neoadjuvant chemotherapy planned

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
