# Peer review of "Beyond N Staging in Breast Cancer: Importance of MRI and Ultrasound-based Imaging"

_cancers, 2022, doi:10.3390/cancers14174270_

Round 1
Reviewer 1 Report
Authors present in this mini review, modalities of axillary node evaluation in breast cancer. They highlight a need for careful evaluation of lymph nodes during loco regional diagnosis of tumor. Authors touch upon basics of axillary staging criteria, anatomy of axilla identifying different node levels using ultrasonography (US), MRI, and histology images. Further, technical details pertaining to operating conditions of US and MRI for optimal evaluation of lymph nodes are provided. In addition, US features of pathological nodes, features of malignancy and morphological evaluation of axillary nodes by MRI for staging of BC is described. Furthermore, authors discuss features of lymph node malignancy and advantages of using MRI based methods for investigation axilla. Authors conclude the review with a discussion about advantages and disadvantages of US and MRI and warrant an integrative approach for optimal axillary node evaluation and how radiomics and AI can bring in value addition to this setting. Overall, the manuscript is highly technical and targeted to clinical audience with useful information on N staging BC.
I have few minor concerns that I have listed below.
The title of the review must be revised. Based on the content of the minireview the appropriate title would be -- Beyond N staging in breast cancer: Importance of MRI and ultrasound-based imaging.
Revise the sentence in simple summary – The N-stage… is fundamental.
Page 8: mention the appropriate reference for the statement – In Baltzer et. al. series ……. Excluding metastasis.
Page 9: Correct Fardanesh et all
Page 9: In the sentence – However, while these differences are significant in this study. Please specify which study. Appropriately revise the paragraph.
Page 10: Please provide images for your description on USPIO-enhanced MRI for evaluation of lymph nodes.
Page 12: Expand more on application of radiomics and AI in N staging of BC.
Author Response
The title of the review must be revised. Based on the content of the minireview the appropriate title would be -- Beyond N staging in breast cancer: Importance of MRI and ultrasound-based imaging.
Corrected as suggested.
Revise the sentence in simple summary – The N-stage… is fundamental.
Done.
Page 8: mention the appropriate reference for the statement – In Baltzer et. al. series ……. Excluding metastasis.
Done.
Page 9: Correct Fardanesh et all
Done.
Page 9: In the sentence – However, while these differences are significant in this study. Please specify which study. Appropriately revise the paragraph.
Specified in the text.
Page 10: Please provide images for your description on USPIO-enhanced MRI for evaluation of lymph nodes.
This is a good idea. Unfortunately, we have not USPIO-enhanced MRI images in our imaging archive because USPIO is not currently used at our Institution because it is not a standardized techinque, For example it is notapproved in USA. This point was added in the text.
Page 12: Expand more on application of radiomics and AI in N staging of BC.
Done.
Reviewer 2 Report
N-staging is important for breast cancer. Imaging plays an essential role by being non-invasive and accurate. This article carefully reviewed imaging methods (ultrasound and MRI) in the staging of lymph nodes for breast cancer. Because treatment decisions will be made based on the N-staging, it is important to achieve high accuracy. The authors started with anatomy of axilla and breast regional lymph nodes. After that, they reviewed the application of ultrasound and MRI, respectively. In the end, they compared these two imaging modalities. In my opinion, this is a well-written review paper. I recommend it to be published in your journal with minor revisions.
1. Fix minor spelling errors
a. Last sentence in the 5th paragraph under “Introduction”
b. “Imaging” above “2. Anatomy of axilla and regional xxxx”
c. First sentence of the last paragraph on Page 9. “Was been developed?”
2. More recent number for breast cancer epidemiology in the first paragraph
3. Confusing sentence under Table 1: “These selected patients showed xxx”
4. Add example images for DWI
Author Response
1. Fix minor spelling error
a. Last sentence in the 5th paragraph under “Introduction”
Corrected.
b. “Imaging” above “2. Anatomy of axilla and regional xxxx”
Corrected.
c. First sentence of the last paragraph on Page 9. “Was been
developed?”
Corrected.
2. More recent number for breast cancer epidemiology in the first paragraph
Added more recent number as suggested.
3. Confusing sentence under Table 1: “These selected patients showed xxx”
Corrected.
4. Add example images for DWI
Added as suggested.